# Role of public involvement in the Royal College of Physicians' Future Hospitals healthcare improvement programme: an evaluation

Lucy Frith, [1] Lauren Hepworth, [1] Victoria Lowers,[1] Frank Joseph,[2] Elizabeth Davies,[3] Mark Gabbay[1]

[1]Institute of Population Health, University of Liverpool, Liverpool, UK
[2]Diabetes, Endocrinology & Internal Medicine, Countess of Chester Hospital, Chester, UK
[3]Royal College of Physicians, London, UK

**Correspondence to**
Dr Lucy Frith;
l.j.frith@liverpool.ac.uk

## ABSTRACT

**Objectives** The Royal College of Physician's (RCP) Future Hospital Programme (FHP) set out a blueprint for a radical new model of care that put patient experience centre stage. This paper reports on the results of an independent evaluation of the FHP and focuses on the role public patient involvement (PPI) played in these projects. The paper explores the perceptions and experiences of those involved in the FHP of how PPI was operationalised in this context, and develops an 'ex-post' programme theory of PPI in the FHP. We conclude by assessing the benefits and challenges of this work.

**Setting** Secondary care. The FHP consisted of eight clinician-led healthcare improvement hospital development sites with two phases.

**Participants** Development site clinical teams, patient representatives, the RCP's Patient and Carer Network, members of the FHP team, and fellows and members of the RCP.

**Design/methods** We conducted an independent evaluation of the FHP using FHP documentation and data collected specifically for the evaluation: qualitative interviews, focus groups and a web-based survey.

**Results** The PPI initiatives set out to develop more patient-centred care and improve the patient experience. The mechanisms designed to meet these goals were (1) a programme of PPI in the development site's projects, (2) a better understanding of patient experience and (3) evaluation of patient experience.

**Conclusion** This evaluation of the FHP identifies some key elements that need to be considered when attempting to more closely integrate PPI and co-production in service re-design. The structure of FHP over two phases enabled learning from phase I to be incorporated into phase II. Having the PPI representatives closely involved, developing communities of practice, and the oversight and measuring activities acted as 'disciplinary structures' that contributed to embedding PPI in the FHP and kept the patient experience at the forefront of the improvement initiatives.

## BACKGROUND

The Future Hospital Programme (FHP) arose out of the Future Hospital Commission (FHC), established by the Royal College of Physicians (RCP) in 2012. The FHC was set

### Strengths and limitations of this study

► Contributes to theoretical developments in public involvement as a quality improvement measure.
► Develops an 'ex-post' theory of how public involvement can be used to improve front-line services.
► This evaluation was unable to get robust data on the effects of the public involvement activities on the development site projects.
► Points to areas for future development such as developing mechanisms for measuring the impact of public involvement on service improvement initiatives.

up in a response to the RCP's *Hospitals on the Edge Report*[1] that found "A health system ill-equipped to cope with the needs of an aging population with increasingly complex clinical, care and support needs; hospitals struggling to cope with an increase in clinical demand; and a systematic failure to deliver coordinated, patient-centred care".[1] The FHC aimed to, 'set out a radical new model of care' that put patient experience centre stage, "Patient experience must be valued as much as clinical effectiveness, and patients must be involved in service design and delivery".[2] A key part of the FHC was to foster and promote patient-centred care, which they defined as "individualised, compassionate, holistic and collaborative… [and] continuous, care in settings appropriate to the patient's clinical and care needs viewed from their perspective".[2] While there is debate in the literature over how this concept is and should be defined,[3] key elements of patient-centred care for the FHC were continuity of care, reduction in unnecessary moving of patients between wards and better integration with social care services post-discharge.

To take the recommendations and vision of the FHC into practice, the RCP set up the

FHP in 2013. The RCP advertised for clinician-led healthcare improvement (HCI) proposals to become FHP development sites. The first phase was recruited in 2014 and focused on improving care for older and frail patients. The second phase was a targeted call on integrated care and began in 2016. Each phase included four pilot project sites; all had to align their objectives with the FHC principles (full details of the FHP have been published elsewhere).[4 5] The programme concluded in 2017, however, the sites continued to work on their quality improvement initiatives beyond the end of the FHP. Applicants were required to:

► Provide details of their projects and how these aligned with the principles of the FHC;
► Demonstrate involvement of patient representatives in design and implementation;
► Have a local, board-level executive sponsor.[4]

The FHP was a multifaceted programme of work that acted as a support mechanism for the sites to improve front-line services, but it did not provide any additional 'transformational' funding. There were four main ways that the FHP supported HCI in the development sites:

*Public patient involvement* (PPI): one of the main elements of the FHP was to foster and support PPI in the development sites' HCI projects working closely with the RCP Patient Carer Network (PCN) (established in 2004 to give greater patient and carer input into the work of the RCP). The aim was to have the patient perspective present throughout the projects and involve patient representatives actively in the co-production of the project design and implementation.

*FHP support structures*: the RCP established a FHP team with a senior physician appointed as Future Hospital Officer for each phase who provided clinical leadership and support. Collaborative learning structures were developed, such as learning events where site teams met and shared experiences, training in improvement methodologies, peer support and creating a community of interest (the Future Hospital Partners Network).[4] The RCP's PCN also provided support to local lay representatives and peer learning opportunities across all sites.

*Reporting processes*: development sites were required to report monthly to the FHP team (both written reports and telephone calls). This ensured that the sites maintained momentum and received regular feedback.

*Data collection and analysis*: sites were given training sessions provided by the RCP or external experts on data analysis, project management and support for collecting and interpreting performance metrics and patient experience data. There was a "focus on measuring the true impact of clinically-led improvement or change… [to] enable clinical teams to improve patient-centred care and outcomes".[4]

This paper reports the results of an independent evaluation of the FHP and focuses on the role public patient involvement (PPI) played. The aims of the paper are twofold: to explore the perceptions and experiences of the central FHP team, the patient representatives and the development site teams of how PPI was operationalised in this context; and to develop an 'ex-post' programme theory of what mechanisms the FHP used to meet its goals of developing more patient-centred care and consequently improving the patient experience. We will conclude by assessing the benefits and challenges of this programme.

The terms PPI and co-production are contested. Tritter defines PPI as "Ways in which patients can draw on their experience and contributors of the public can apply their priorities to the evaluation, development, organisation and delivery of health services".[6] When defining the co-production, it is important to recognise that, in the messy realities of practice, how patients and the public participated in the FHP did not always conform to one definitional model, but often included and straddled different types of practice: from passive involvement, such as being consulted about planned changes, to more active involvement, such as being involved in planning; and, in some cases, co-production of initiatives.[4 5] Co-production was used in FHP to mean, "equal partnerships between patients and physicians in the design of health services".[4] Osborne *et al*[7] define co-production as "the voluntary or involuntary involvement of public service users in any of the design, management, delivery and/or evaluation of public services". As has been noted, the concept of co-production is very broad and specifically, for the FHP, the key element of co-production was service co-design,[8] and we shall use the term co-production/design to denote the form co-production took in the FHP. Definitions of healthcare improvement are also contested.[9] We will define HCI broadly as "better patient experience and outcomes achieved through changing provider behaviour and organisation through using a systematic change method and strategies".[10]

There is a growing literature on the role of PPI in HCI,[6–8] but this remains underexplored compared with the use of PPI in other areas, and this paper contributes to this developing area. There is a large literature on theoretical considerations and topologies of involvement and a number of attempts to develop categorisations to capture what PPI is[6 11 12] and what an 'ideal type' might be.[13] Arnsteins' ladder of engagement is often quoted as one of the first examples of this, where the bottom rung is manipulation and the top rung is citizens' power.[14] This paper does not attempt to compare or measure PPI in the FHP according to abstract standards, but contributes to the empirical literature on the practice of PPI in service re/design and quality improvement. It has been noted that although there is a substantial body of theoretical literature, there is a gap on *how* involvement and co-production/design is carried out in practice.[15] It is widely recognised that the context of interventions are key,[16] and this paper provides an example of a specific programme of PPI to add to the literature on exploring contextual factors and developing good practice in the area of PPI in HCI.

## METHODS

This paper is based on a summative evaluation[17] our team conducted of the FHP as a programme of HCI. It does not consider the specific site improvement projects nor content of these changes (these are reported elsewhere).[4 18–25] This paper reports on the perceptions and experiences of the central FHP team, the patient representatives and the development site teams, of how PPI was operationalised in this context and the benefits and challenges of this work. Our team became involved towards the end of the programme and we did not have input in the design of the projects, nor the procedures, content and timing for gathering metrics.The challenges in doing this kind of research have been discussed in the literature[26] and this paper reports on what happened in a HCI initiative in the real world, not one that took place as part of a structured research project. There are a number of strengths and limitations with using evaluations as research data. The strengths are that it provides information on what actually happened in practice and it demonstrates the workings of PPI embedded in the context of shifting priorities and changes in the NHS, such as funding and personnel changes. The main weakness is that from the data collected by the FHP and our evaluation, it was difficult to ascertain the effects of the PPI initiatives on the individual development site projects.

We used a multisource approach using a variety of data sources. We analysed data the FHP collected, reviewing 471 documents from the development sites, including monthly, quarterly and annual reports, patient reports, learning event presentations, posters, feedback, monthly call notes, notes from site visits and personal communications. Members of the team also attended some of the national meetings for the development sites.

We also collected data specifically for the evaluation: focus groups, qualitative interviews and a web-based survey. Focus groups were conducted with all the development site teams, which included the clinical lead and at least one patient representative. A further focus group was conducted with seven local patient representatives and two PCN representatives, in which seven of the eight development sites were represented. One-to-one interviews were offered to all local patient representatives, four were conducted via telephone and one patient representative provided a written account. One-to-one interviews were conducted with key personnel in the RCP, 17 individuals either face to face or via telephone, six from within the FHP core team and 11 who were indirectly involved with the FHP, including three senior members (see table 1 for an overview).

A web-based questionnaire was circulated to the development site team members including the patient representatives, providing another opportunity for team members to share their opinions individually. The survey used a combination of multiple choice and free-text response options, with questions based on the topic guide for the focus groups. The survey received 22 responses; this included at least one response from each of the eight

| Table 1 | Overview of qualitative data | |
| --- | --- | --- |
| | **Number** | **Participants** |
| Focus group | 1 | 9 (7 site patient representatives, 2 PCN members) |
| | 8 | With each development site team. Between 2 and 5 team members. All groups included the clinical lead and 5 out of the 8 groups included at least one patient representative |
| Interviews | 5 | Patient representatives |
| | 6 | FHP core team |
| | 11 | Members of RCP (including 3 senior members) |

FHP, Future Hospitals Programme; PCN, Patient Carer Network; RCP, Royal College of Physicians.

development sites and from a mixture of patient representatives and clinicians. (for details see Hepworth *et al*[5]). The full data and analysis are available in the evaluation report,[5] evaluations of the individual site projects[18–25] and the final FHP Report.[4]

All focus groups and interviews were recorded with consent and transcribed verbatim, and a constant comparative thematic analysis was used to code the data.[27] Descriptive quantitative statistics were used to analyse the multiple-choice responses of the survey. To maintain participants' anonymity, development sites are referred to as DS1 to DS8, patient representatives as PR1 to PR6 and the personnel in the FHP team as RCP1 to RCP17 (see table 2 for information on the participants who we quoted). The manuscript was prepared using the Standards for Quality Improvement Reporting Excellence (SQUIRE 2.0).[28]

## RESULTS

The FHC saw 'patient experience as important as clinical effectiveness' and that patient-centred care should drive the improvement of services.[2] This commitment underpinned the FHP's attempt to re-orientate hospital provision to be more responsive to patients' needs rather than organised around professional and organisational imperatives.[4] The FHP aimed to do this by consulting with patients and the public and also initiating processes and structures to facilitate greater PPI and co-production/design of initiatives in the development sites' HCI projects.

As has often been noted in the literature,[29] HCI programmes seldom start with an explicit programme theory, defined as "an explanation of why the effects

**Table 2** Participant information for those quoted in the paper

| ID | Phase | Site | Role |
|----|-------|------|------|
| PR1 | I | 5 | Patient representative |
| PR2 | I | 5 | Patient representative |
| PR3 | I | 4 | Patient representative |
| PR4 | I | 3 | Patient representative |
| PR5 | II | 6 | Patient representative |
| PR6 | II | 7 | Patient representative |
| DS3 | I | 3 | Clinician |
| DS8 | II | 8 | Clinician |
| RCP1 | – | – | FHP team |
| RCP4 | – | – | FHP team |
| RCP5 | – | – | Other RCP employee |
| RCP6 | – | – | FHP team |
| RCP12 | – | – | FHP team |

FHP, Future Hospital Programme; RCP, Royal College of Physicians.

observed in a programme are likely to have occurred".[30] Simply describing interventions does not always ensure that they can be replicated or scaled up in other contexts. How certain outcomes come about, the underpinning social processes and the mechanisms that produce these outcomes need theoretical interpretation to aid replication of the intervention.[31] Developing theory can also form the basis of 'theory orientated evaluation', which aims to identify "the rationale and assumptions about mechanisms that link programs' processes and inputs to outcomes (both intended and unintended)".[30] In order to conduct a form of theory-orientated evaluation, we developed a programme theory of PPI as a HCI intervention in the FHP. As the FHP did not set out an explicit programme theory at the outset, we constructed a form of 'ex-post theory'.[30] We developed this 'ex-post' programme theory from our analysis of the data we gathered as part of the evaluation and the wider FHP documentation.

The main goals of the PPI procedures in the FHP were to develop more patient-centred care and consequently improve the patient experience. The mechanisms designed to meet these goals can be broken down into three elements:

1. Greater PPI and co-production/design of the initiatives: having PPI representatives closely involved in the development site teams will keep them focused on developing patient-centred care and improving the patient experience, so that this is less likely to be overlooked or marginalised.
2. A better understanding of patient experience: organisations and professionals may not know what constitutes 'good patient experience'. PPI can provide important, hitherto, missing information and perspectives on pa-

tient experience that can be fed into service redesign alongside operational and clinical information.
3. Evaluating patient experience: using patient experience data as a key metric for evaluating quality of care will re-orientate clinical teams to the importance of patient experience and aid the prioritisation of patient-centred care.

How these mechanisms were operationalised in practice and determining the successful aspects of the FHP are important for thinking through how PPI can be used in service change and improvement and what lessons can be learnt for future HCI projects. We have presented the results to address each of these three mechanisms.

## PPI and co-production/design of initiatives

Having PPI representatives closely involved in the development site teams was seen as a mechanism to begin to involve patients and move towards greater co-production/design of services to ensure the continual prioritisation of patient-centred care. PPI was implemented from the start of the FHP. The RCP's PCN was involved in the FHC and setting up the FHP. Each development site had at least one local PPI representative within their team, and in phase II, sites had to show that they had a PPI strategy in place before being chosen. The PPI representatives were meant to be an integral part of the FHP team and be active participants in team meetings and decision-making processes. The majority (63.6%) of respondents to the development site web-based survey reported that their project was partially co-produced/designed with patients and 27.3% reported that their projects fully involved patients. Co-production/design was limited in some sites, particularly in phase I, where some projects were already fairly well developed before any PPI took place.

Although in each phase the development sites had the same remit and operating instructions, it was clear that PPI did not happen in the same way and to the same degree in each site. Comparing the evidence across site reports and phases, there were different levels of involvement and the timings and format of PPI varied. In some instances, the involvement was reported to be rather 'tokenistic' and the patient representative's chance to influence service redesign was seen as limited (table 3, quote 1).

In some of the sites, the PPI representative had little input into setting up the project or its design, but were more involved at later stages. Some participants felt they were brought in after the main decisions were taken (table 3, quotes 2 and 3). However, in some sites, patient representatives were more actively involved in setting up and co-designing the projects from their inception (table 3, quote 4).

The FHP central team reported that from their perspective, PPI was well embedded in the FHP and this enabled more co-production/design of projects (table 3, quote 5).

**Table 3** Levels of involvement and co-production/design of initiatives

| | **Involvement and co-production/design of initiatives** |
|---|---|
| Quote 1 | "Well I have done very little. I have, I haven't had no idea what the patient rep was supposed to do. In my opinion I was merely a tick in a box that said you have to have a patient rep". (PR4) |
| Quote 2 | "It quite frankly is that you don't start off with asking the patients what they want, you start off usually with some enthusiastic usually a clinician, who has an idea about how things might be done better… and the patients are asked to contribute to the development of that idea". (PR5) |
| Quote 3 | "All you ever do is ask them to review what you have done rather than to input into it and you know there are these things where you go, hmm, this is not a co-production the patient is not at the heart of the process of the project". (PR6) |
| Quote 4 | "(I was) very involved. We meet monthly with the… team which is an opportunity to share and discuss ongoing proposals and ideas or implementation of new approaches to working…. The patient reps are treated with courtesy and respect and views are listened to and taken on board. We are considered to be an integral and vitally important part of the team". (PR2) |
| Quote 5 | "Unique to Future Hospital, in comparison to the other programmes of work. So there is lots of 'oh a patient was involved' tick type activity, throughout the College (RCP), and I think the difference particularly with the phase two sites is that there is proper co-production with the patients I hope that they feel that way, it certainly seems at least a big step along the route to co-production, than anything else that I have been involved with or seen or heard about so far". (RCP1) |

### Better understanding of patient experience

The FHP sought to get a better understanding of patient experience in two main ways. First, the involvement of the PPI representatives in the site teams; and second, the capture of wider patient experience data in each development site. These will be discussed in turn.

The involvement of the PPI representatives in the site teams was seen as important by members of the central FHP team as a way of keeping the focus of the development site teams on patient experience (table 4, quote 1).

However, the role of the PPI representative, what it should be and who was an appropriate person to occupy this type of role were debated by all FHP participants, and there was little consensus on this. The key area of debate was who could be said to accurately provide insights into the patient experience, and how do we know we are not just getting a partial view or views that may not be held by the majority of patients? As one site noted: "those that are able to meaningfully contribute to system redesign are not necessarily representative of the wider population".[4] Underpinning these concerns was ambiguity over the concept and purpose of being a 'representative' and what 'representativeness' meant in practice (table 4, quotes 2 and 3).

A clinician in a development site, while supportive of having patients involved, questioned the utility of having a specific PPI representative, as they felt this just reflected one

**Table 4** Better understanding of patient experience

| | |
|---|---|
| Quote 1 | "It has helped to keep the clinicians grounded, it has helped to keep the focus on patient experience". (RCP12) |
| Quote 2 | "I am very much of the opinion that individual patients cannot represent patients as a whole unless it is very strange or peculiar circumstances. Probably, leaders of some patient organisation or something but even then it is a pretty poor sample". (PR5) |
| Quote 3 | "There is still quite a lot of uncertainty about what your (PPI representative) role is. Are you giving a viewpoint as a patient who has experienced that service, so if you like common sense from an individual point of view or are you in a representative role are you trying to reflect a broader view of patients let's say who are acutely ill going in through a particular hospital. And, that I think hasn't been worked out nationally we haven't really got a sort of sense of what the, what the major aspects of a patient representative role are". (RCP12) |
| Quote 4 | "I think that the disappointing thing is that one considers patient experience, to be reflected by a patient representative. Because patient experience is so much more than just one person coming in and saying… I think the word representative is a very difficult word because I am not sure that (our patient representative) could truly represent patients other than having been one… we have 70 000 patients a year, so the question in my mind is, you know if we are trying to extrapolate a representation of their experience, then having one person who has their own carried experience representing them is difficult". (DS3) |
| Quote 5 | "If you have a patient representative the… are they a representative of the wider population or are they just bringing their own baggage to it, that is a big question and it is a big, you almost have to train people not to bring their baggage to the table and that is not easy". (RCP4) |
| Quote 6 | "Patients will speak more openly with us (PPI representatives) than perhaps they feel they can do with the medical or other members of the team". (PR2) and without their input, "I don't believe their true voice would have been heard. And so some assumptions would have been made as to what the patient needs". (PR2) |

individual's opinion (table 4, quote 4). There was a concern over the lack of capacity in PPI and one member of the central FHP team felt that training was needed to enable PPI representatives to step outside of their own concerns and be able to represent the wider group (table 4, quote 5).

A second way of incorporating data on patient experience in the FHP was the systematic capture of patient experience data in each development site.[30] Local patient and PCN representatives played a key role in organising and developing patient experience surveys and other types of data gathering such as running focus groups or public events. The PPI representatives were involved in recruiting, training and supporting volunteers to pilot the patient experience surveys, provide feedback, propose improvements and develop site-specific tailored questionnaires.[32] These were seen as one of the successes of the FHP, that they were able to get the feedback from the wider body of patients and in a more sustained and nuanced way. One development site patient representative spoke about how they were able to get much richer and valuable data on patient experience and what patients wanted from services (see table 4, quote 6).

### Metrics on patient experience

An important part of the FHP was building capacity in data collection and its management so that the sites could demonstrate the impact of their service changes. One of the key elements of this was developing expertise in capturing, measuring and interpreting patient experience data. "This metrics orientation provides focus and develops a language of 'measures' that 'quantifies' patient care".[4] It was seen as important that the metrics captured reflected what was important to patients.[4] The role of this continual data collection in prioritising PPI was seen as crucial to the success of instituting PPI in the FHP (table 5, quote 1).

One of the benefits of this continual reporting and assessment of progress and the incorporation of a form of the 'Plan-Do-Study-Act' (PDSA) cycle[33] was that the FHP's approach to PPI evolved between the two phases. This was an iterative process where lessons were learnt and improvements made as the programme progressed. Due to problems experienced in phase I, where some sites lacked any patient representation for periods of time or had recruited individuals who had difficulties

in contributing to the role, changes were made to the application process to be a development site. In phase II, "Patient representatives were identified in the application and were integral to the interview processes".[4] The PCN became involved in the selection of the sites and the recruitment for the local site PPI representatives became more formalised. The development of the PPI representatives' role and the overall strategy and implementation over the two phases of the FHP is outlined by a member of the FHP team (see table 5, quote 2). Other improvements included providing more support for the development site PPI representatives. For example, they were given a buddy from the central PCN, and this relationship became more formalised in phase II.

### DISCUSSION

The FHP commitment to involving local patients, PPI representatives and PCN members in service design, monitoring and evaluation was a new experience for the RCP and the PCN.[32] It was recognised by all parties that 'doing' PPI was challenging and required significant commitment on the part of all members of the development site teams.

The lack of theoretical consideration of how PPI might contribute to service change and create the conditions for patient-centred care can make evaluation challenging, and the paucity of underpinning theory in this area has been noted in the literature.[29 30] In order to design, carry out and continuously evaluate PPI in HCI projects, having a working programme theory can provide clear aims and objectives against which to assess whether the PPI mechanisms have fulfilled their stated aims and if, and how, PPI as an intervention has worked. How such theoretical underpinnings of PPI as an intervention can and should be developed are areas that need further investigation. We developed an 'ex-post programme theory' of PPI as an HCI intervention and setting out the goals for this kind of programme is a first step to developing ways of assessing these initiatives. Our study has shown that more work needs to be done on developing appropriate metrics for assessing PPI as a HCI tool.

The initial requirements for sites to have a PPI strategy, a patient representative and good links with the PCN were important framing mechanisms for the project. This kept the focus on patient experience and it was by being a FHP development site that this was promoted and sustained. As

| Table 5 | Metrics on patient experience |
|---|---|
| Quote 1 | "… the patient engagement piece is quite labour intensive, and whether without the Future Hospital Programme guiding it… will providers prioritise it (patient engagement) in the same way they would without the Future Hospital chasing them and asking them what they are doing on a regular basis…". (DS8) |
| Quote 2 | "For phase two having watched how the sort of organic coming together, people volunteering option in phase one for patient reps, hadn't really been as effective as I think we initially hoped and that, the rigours of doing FHP alongside the day jobs… meant that patient recruitment was often low down on their list and sometimes it took them, a good few months to recruit a patient rep…. So for phase two we decided to do a more proactive recruitment campaign advertised… So we learnt a lot from phase one, about how to take patient involvement and engagement in the development site teams from sort of tokenistic and leaving the teams to do it themselves, to really prescribing what we needed and to get the framework in place to". (RCP6) |

site members observed, along with all the other requirements of both the HCI projects and day-to-day business, it would have been easy for PPI to get subsumed. The communities of learning that the FHP support structures created and attending learning events with other sites[4] acted as a disciplinary structure, a community of practice that kept PPI at the forefront of the projects.[30] The support and endorsement for PPI from a prestigious medical college was also a key factor in PPI being seen as an important and worthwhile activity. This way of improving practice through professional communities and expert reassurance has been used in other studies,[34] and worked well in the FHP, although other studies suggest more mixed evidence of their effectiveness.[35]

A challenge with getting comprehensive information on patient experience is that only a few patients were and could be involved in the PPI initiatives, and therefore, this information could be argued to only partially capture the patient experience. Even when sites tried to organise public involvement days, these often did not attract much interest from patients. Further, concepts of how patient representatives 'represent' are contested in PPI.[36] Several sites saw the role of the patient representatives as a 'patient voice' rather than providing a representative account of 'patient experience'. Hence, the ability to attract a range of patients' views and get wider understanding of patient experience was often limited. However, sites generally saw the value of including patient representatives in clinical teams and reported that they found the contribution of patients extremely valuable. Several sites were aiming to increase the level of patient input, to help them respond to the high levels of changing demand among their population groups. Thus, adding extra in-house capacity and developing arrangements for closer working with patient groups was seen as a high priority for future service improvement. The work done by the patient representatives—running patient surveys, developing and piloting new iterations, and exploring qualitative data gathering mechanisms, such as interviews and focus groups—helped produce much richer information, and site teams used this to both monitor their progress and feed into further improvement cycles.

Finally, the FHP was designed to enable sites to instigate robust systems for measuring progress and giving the sites the skills to set up data gathering structures. 'Measurement always has consequences',[30] and by having patient experience data as a key performance indicator and the requirement to report to the central FHP team regularly, it ensured both that this remained an important focus for the projects, and that the patient representatives had a clearly defined and important role.

### Learning between phases

A strength of the FHP was employing learning from phase I into phase II and embedding a PDSA cycle in the programme. This meant key changes were made in phase II that strengthened PPI: involving the PCN in the process of choosing sites, strengthening processes for recruiting patient representatives, formalising the buddy system between the local representatives and a PCN member, and recommending each team have two patient representatives to share the work and provide support.[32]

### Limitations

This was an evaluation[5] of an existing and ongoing project, and the data and measurement processes were not determined by the evaluation team. Thus, this evaluation is based on the available data and resources. The FHP did not include an impact assessment of the PPI activity itself and hence, we were not able to get detailed process or quantitative data on the activities undertaken by either the local or PCN patient representatives.[5] As has been noted, assessing the impact of PPI on HCI projects is challenging, and embedding key measures of success throughout the project would have provided a useful way of assessing progress during the FHP. The lack of studies on how PPI effects the outcomes of service reconfigurations has been noted in Dalton *et al*'s review,[37] and ways to measure the impact of PPI are generally under-researched.[38] This is an area that needs further research to develop ways of effectively capturing the impact and effects of PPI in service design and change. For future HCI, our study highlighted the need for improvement teams to have a programme theory of how the intervention could work from the offset that includes clear goals of the initiative, how concepts such as co-production and related concepts such as co-design are defined and subsequently employed, what markers of success might look like and clear mechanisms for achieving this. By having a programme theory to guide setting up the initiative and establishing mechanisms of achieving key outcome measures, it should be possible to determine the impact of PPI in HCI with more accuracy.

### CONCLUSIONS

Developing an ethos of co-production in the form of co-design, even if this was often moving towards rather than fully achieved, was a major success of the FHP. To meaningfully embed PPI in HCI projects, it needs to be sustained and sustainable, and requires significant investment in support structures, both centrally and locally, plus time and space allocated to enabling decisions to be genuinely co-produced. There needs to be genuine support from the whole system—organisational buy-in—so that it becomes a 'way of doing things' throughout the HCI project and once the initiative is rolled out. Additionally, there needs to be proper investment of staff time and associated costs to deliver real PPI and co-production. For example, the project teams need dedicated time for meetings to start the design process, rather than involving patient representatives when the projects are already set up, as this inevitably precludes genuine co-production. Demonstrating the impact of PPI in HCI projects is the next challenge and there is a need to develop methodologies that can capture relevant metrics and the nuances of involvement. Combining qualitative and quantitative

data and embedding collection throughout the project can give a more multidimensional picture.[26] Attention to developing programme theories of PPI in HCI will be a key step, and this paper aimed to contribute to this debate by developing an 'ex-post theory' that could be used to suggest ways of developing PPI as an HCI tool.

**Acknowledgements** We would like to acknowledge and express thanks to all the members of the development site teams, the PCN, staff at the Royal College and Physicians and Members and Fellows of the RCP who took the time to speak to us or complete a survey. The authors would also like to thank Professor Tom Walley for his support with this work. Professor Mark Gabbay and Dr Lucy Frith are part-funded by the National Institute for Health Research (NIHR) Collaboration for Leadership in Applied Health Research and Care North West Coast (CLAHRC NWC).

**Contributors** LF: analysed the data, drafted the paper. LH, MG, VL: analysed the data, contributed to writing and revising the paper. FJ: contributed to writing and revising the paper. ED: public contributor, contributed to writing and revising the paper.

**Funding** This evaluation was funded by the Royal College of Physicians.

**Disclaimer** The views expressed here are those of the authors and not necessarily those of the NHS, the NIHR, or the Department of Health and Social Care.

**Competing interests** None declared.

**Patient consent for publication** Not required.

**Ethics approval** Ethical approval was given by the evaluation team's Institution.

**Provenance and peer review** Not commissioned; externally peer reviewed.

**Data availability statement** Data are available on reasonable request.

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
