## [Reviewer comments · BMJ Open]

ARTICLE DETAILS

TITLE (PROVISIONAL)	The role of public involvement in the Royal College of Physicians' Future Hospitals healthcare improvement programme: an evaluation.
AUTHORS	Frith, Lucy; Hepworth, Lauren; Lowers, Victoria; Joseph, Frank; Davies, Elizabeth; Gabbay, Mark

VERSION 1 – REVIEW

REVIEWER	Helen Barratt University College London, UK
REVIEW RETURNED	17-Jan-2019

GENERAL COMMENTS	Thank you for the opportunity to review this paper, which seeks to address an interesting and important topic. As others have noted, there is a real need for independent evaluations of public involvement in service change, particularly regarding its impact (see Dalton et al, JHSRP 2016; 21(3):195–205 for example). My feedback is as follows: 1) It would be helpful to have more detail in the introduction about the Future Health Programme. In particular: How many sites were recruited? How long did the programme run? What types of improvement initiative were eligible to apply (beyond 'care for older and frail patients' and 'integrated care')? What was the scale of these? How did the FHP 'foster and support PPI'? Who were the Future Hospital Officers, and how did their role work? What does 'report regularly' mean? All this is important context to help the reader understand how the programme was operationalised.2) Also in the introduction, it would be useful to have key terms defined. These include PPI, health care improvement, and co-production. Making clear the distinction between PPI and co-production throughout the paper is particularly important. At times the authors seem to conflate the two concepts. For example, the third paragraph of the discussion (line 322 onwards) describes PPI generally, rather than actual co-production.3) Linked to this, the authors make only brief reference to the large literature – both empirical and theoretical – on PPI in service improvement and co-production. Situating their work more explicitly within this would increase the richness of their analysis.4) I am not completely certain what the aim of the paper is, and what the specific objectives/ research questions were. Several goals are mentioned in the lines 120-129. However, I'm not sure that these are all fully addressed. For example, the paper doesn't seem to analyse in depth 'how the FHP used PPI to drive change
---

and improve patient experience', nor does it really analyse the 'underlying theoretical assumptions for embedding PPI in the FHP.' On this latter point, the authors develop their own programme theory of the intervention. I'm also not clear from the results 'how PPI contributed to the development of the individual projects.' See comments below on the results section.

5) The web-based questionnaire is only mentioned briefly in the methods section. What did it contain, how was it developed, and how did it contribute to this analysis?

6) Throughout the paper, it would be helpful if the language were more precise. For example, 'sites were given sessions' (line 114), 'attended some of the national meetings' (line 137), 'other types of data gathering' (line 273)

7) I am not certain how the authors developed their programme theory, and how much of this was drawn from FHP documentation, non-participant observations etc. Did they explore the rationale behind the approach at all in the interviews/ focus groups?

8) I have a number of concerns about the results section, particularly the extent to which it describes an examination of the programme theory:

- In terms of the overall structure of the results, see my comments above about the aims/ research questions.
- Lines 322-359 of the discussion contain quite a bit of new information/ data, which is mentioned here for the first time. This would be better in the results section.
- The first sub-section is focused on co-production, working with the rationale that 'having PPI representatives closely involved in the development sites teams will keep them focused on patient experience and patient-centred care.' Co-production to me implies that patient representatives are equal partners and co-creators of service change (ie power is shared to an extent). Involvement, on the other hand, is a much broader concept, and could be entirely passive. The results section and associated discussion text seem to focus mostly on the extent to which there was involvement, rather than actual co-production at each of the sites.
- The second sub-section looks at understanding patient experience, with the rationale that PPI can provide information about patient experience to feed into service redesign. The focus, however, is largely on the extent to which the individuals involved could be considered to be representative, rather than how/ whether patient experience data were used to drive change.
- The third sub-section looks at patient experience metrics, on the basis that 'using patient experience data... will orientate clinical teams to the importance of patient experience.' The first paragraph (line 283 onwards) focuses on processes to capture patient experience metrics. The second paragraph describes changes in the emphasis placed on PPI as the programme evolved. I can't see from the text, here or in the discussion, how patient experience data were used by the clinical teams, or indeed how PPI in general impacted the work of the FHP sites, beyond it being an important focus (line 358).
- In summary, the authors state in the discussion that they have used the programme theory to assess whether the PPI mechanisms had fulfilled their aims. I'm not certain from the text whether this was the case. In particular, I am unsure whether and how PPI impacted service change at the study sites.

REVIEWER	Simon Malfait Ghent University Hospital
REVIEW RETURNED	26-Feb-2019

GENERAL COMMENTS	First, the authors should be congratulated on choosing the approach of PPI in redesigning the provision of healthcare. Patient and Public Involvement is indeed a suitable method to put patient experience at the center of attention within healthcare institutions. As literature shows, there is need for more evidence-based knowledge on this topic and the authors try to improve this body of knowledge. There are however, in my opinion, apparently several (methodological) flaws in the paper. Whether these flaws are a result of actual limitations of the study or a focus on the wrong elements while writing the manuscript is unclear to me. Therefore, my advice is to provide the opportunity for the authors to revise their manuscript and provide more clarity on these issues. If the elements mentioned below are indeed methodological flaws (and can't therefore be amended), I would advise the authors to rewrite their manuscript as a discussion paper, rather than an actual research paper. Again, some insights in this paper could be valuable for future initiatives concerning PPI, even if these are based on experiences instead of rigorous research. Background: Although it is useful to receive necessary information on the Future Hospital Program, more information and literature on theoretical models and insights on PPI and co-creation is missing. In the past years many research articles have informed practice concerning these topics. Without referring to international examples of PPI, the authors create the idea that their initiative lacks theoretical input and diminishes transferability. Amongst other examples, I advise the authors to look at initiatives from the U.K., The Netherlands, Germany, France and Belgium. Also, concerning co-creation, a paper by Castro et al. was recently published in Patient Education and Counseling. It would also be advisable to make a clear distinction between patient-centred care and patient experiences. I advise the authors to look at the editorial of Dawing and McCormack from 2017. Overall, a more theoretical input is needed in the background section. Methods: The authors state in line 132 that they use a mixed-methods approach to analyze the data. Yet, when reading the paper, it seems that the authors simply have used different methods. Mixed-methods research implies that results from methods are integrated and used in a specific order. I advise to take a close look at: Johnson, R. B., Onwuegbuzie, A. J., & Turner, L. A. (2007). Toward a definition of mixed methods research. Journal of mixed methods research, 1(2), 112-133. Moreover, it seems that the quantitative part is missing/not executed in the paper. If so, there is no need of mentioning this in the abstract. I'm also missing information on ethical consent and ethical approval of this study. Results: it is unclear to me which information comes from the analyzed documents, the interviews and are interpretations of the researchers. The result section appears very chaotic and is, at some points, very difficult to follow. In order to provide more structure and make the results more objective, a clearer referral to the data sources is needed. Moreover, it seems strange to me that at this point of the paper, new references are added. What is their
--

	goal? If they endorse some statements made, this should be discussed in the discussion section. Also, Table 2 is unnecessary (no added value). Discussion: The discussion section is mostly a reproduction of the results and is writing in a very setting-specific manner. Due to this type of writing, it is difficult to extract insight that can be useful for other initiatives in other countries. My advice is to make the findings more abstract to make them transferrable to other settings. This could be done by crosschecking the finding in the paper with international models on PPI. This would enhance the strength and importance of the paper. Limitations: In my opinion, one of the major limitations is the underrepresentation of PPI-stakeholders throughout the process (i.e. focusgroups and interviews). As research has shown before, such underrepresentation affects the role that PPI-stakeholders can/are willing to play and the input they deliver. Please elaborate on this issue. Furthermore, the limitations sections is relatively short in comparison to the other parts of the paper whilst many flaw in the research methodology are present. Conclusion: Based on the comments above, the conclusion will have to be rewritten. I do advice the other to retain phrase 377-to 378 and make this the center of the conclusion. Reading this sentence suddenly gave more meaning to the whole findings and study.
--	---

VERSION 1 – AUTHOR RESPONSE

Reviewer(s)' Comments to Author:

Reviewer: 1

Reviewer Name: Helen Barratt

Institution and Country: University College London, UK Please state any competing interests or state 'None declared': None declared

Please leave your comments for the authors below Thank you for the opportunity to review this paper, which seeks to address an interesting and important topic. As others have noted, there is a real need for independent evaluations of public involvement in service change, particularly regarding its impact (see Dalton et al, JHSRP 2016; 21(3):195–205 for example). My feedback is as follows:

1) It would be helpful to have more detail in the introduction about the Future Health Programme. In particular: How many sites were recruited? How long did the programme run? What types of improvement initiative were eligible to apply (beyond 'care for older and frail patients' and 'integrated care')? What was the scale of these? How did the FHP 'foster and support PPI'? Who were the Future Hospital Officers, and how did their role work? What does 'report regularly' mean? All this is important context to help the reader understand how the programme was operationalised.

Response: we have added in more contextual detail on the FHP, throughout the background section. We have also noted that full details of the FHP and our evaluation have been published elsewhere and are available in open access formats on the RCP webpages.

2) Also in the introduction, it would be useful to have key terms defined. These include PPI, health care improvement, and co-production. Making clear the distinction between PPI and co-production throughout the paper is particularly important.

Response: we have clarified how we are using these terms by adding definitions on pages 3, 5-6 and be consistent in the use of co-production throughout the paper.

At times the authors seem to conflate the two concepts. For example, the third paragraph of the discussion (line 322 onwards) describes PPI generally, rather than actual co-production.

Response: we have clarified this point, an aim of the FHP was co-production of change, but overall this was often 'moving towards' rather than achieving this – so we have clarified this in the results and the discussion.

3) Linked to this, the authors make only brief reference to the large literature – both empirical and theoretical – on PPI in service improvement and co-production. Situating their work more explicitly within this would increase the richness of their analysis.

Response: we have added in some additional literature and situated our study in the literature on empirical work on what works in specific contexts for PPI in HCI) (page 6-7).

4) I am not completely certain what the aim of the paper is, and what the specific objectives/ research questions were. Several goals are mentioned in the lines 120-129. However, I'm not sure that these are all fully addressed. For example, the paper doesn't seem to analyse in depth 'how the FHP used PPI to drive change and improve patient experience', nor does it really analyse the 'underlying theoretical assumptions for embedding PPI in the FHP.' On this latter point, the authors develop their own programme theory of the intervention. I'm also not clear from the results 'how PPI contributed to the development of the individual projects.' See comments below on the results section.

Response: We would like to thank the reviewer for their comments. This paper presents the result of an evaluation that we carried out towards the end and after the FHP had been completed. We have clarified the aims and scope of the paper on pages 5-6 & 7, making it clear that this was an evaluation of the FHP and, therefore the aims of this paper, reporting the evaluation have been clarified. Further, what data and metrics were available to us limited our ability to ascertain the impacts of the PPI initiatives in the development sites.

5) The web-based questionnaire is only mentioned briefly in the methods section. What did it contain, how was it developed, and how did it contribute to this analysis?

Response:

We have added details on the survey in the methods on page 8. It contributed to the analysis in providing additional views on the FHP and we have mentioned some of the relevant findings from this in the results (page 11).

6) Throughout the paper, it would be helpful if the language were more precise. For example, 'sites were given sessions' (line 114), 'attended some of the national meetings' (line 137), 'other types of data gathering' (line 273)

Response: we have added precise details, where available, throughout the paper.

7) I am not certain how the authors developed their programme theory, and how much of this was drawn from FHP documentation, non-participant observations etc. Did they explore the rationale behind the approach at all in the interviews/ focus groups?

Response: we have clarified how we developed this theory and its relationship to the data we gathered. We did not discuss the theory development in the data gathering phase as the theory arose out of the data we gathered and the wider FHP documentation.

I have a number of concerns about the results section, particularly the extent to which it describes an examination of the programme theory:

Response: see response to point 7.

8. Lines 322-359 of the discussion contain quite a bit of new information/ data, which is mentioned here for the first time. This would be better in the results section.

Response: these refer to general features of the FHP that were outlined when giving an overview of the FHP, rather than specifically in the results, we have re-written the discussion at page 17 to show that these are reflections on the programme.

9. The first sub-section is focused on co-production, working with the rationale that 'having PPI representatives closely involved in the development sites teams will keep them focused on patient experience and patient-centred care.' Co-production to me implies that patient representatives are equal partners and co-creators of service change (ie power is shared to an extent). Involvement, on the other hand, is a much broader concept, and could be entirely passive. The results section and associated discussion text seem to focus mostly on the extent to which there was involvement, rather than actual co-production at each of the sites.

Response: we have clarified this point, an aim of the FHP was co-production of change, but overall this was often 'moving towards' rather than achieving this – so we have clarified this in the results and the discussion. As mentioned above, we have also included definitions of these key terms to make the distinction between involvement and coproduction clearer.

10. The second sub-section looks at understanding patient experience, with the rationale that PPI can provide information about patient experience to feed into service redesign. The focus, however, is largely on the extent to which the individuals involved could be considered to be representative, rather than how/ whether patient experience data were used to drive change.

Response: we have clarified this, at page 13, the point here is that there was debate over who should be involved to be able to accurately reflect the patient experience. We have made the aims of the paper clearer to also address this i.e. our evaluation did not allow us to get the data or metrics on how far patient experience data had driven change in any clearly quantifiable way.

11. The third sub-section looks at patient experience metrics, on the basis that 'using patient experience data... will orientate clinical teams to the importance of patient experience.' The first paragraph (line 283 onwards) focuses on processes to capture patient experience metrics. The second paragraph describes changes in the emphasis placed on PPI as the programme evolved. I can't see from the text, here or in the discussion, how patient experience data were used by the clinical teams, or indeed how PPI in general impacted the work of the FHP sites, beyond it being an important focus (line 358).

Response: again, this relates to the overall aim of the paper, and the type of data we are presenting, see response to point 10.

12. In summary, the authors state in the discussion that they have used the programme theory to assess whether the PPI mechanisms had fulfilled their aims. I'm not certain from the text whether this was the case. In particular, I am unsure whether and how PPI impacted service change at the study sites.

Response: (as response to point 4) This paper presents the result of an evaluation that we carried out towards the end and after the FHP had been completed. We have clarified the aims and scope of the paper on pages 5-6 & 7, making it clear that this was an evaluation of the FHP and therefore what data and metrics were available to us limited our ability to ascertain the impacts of the PPI initiatives in the development sites.

Reviewer: 2

Reviewer Name: Simon Malfait

Institution and Country: Ghent University Hospital Please state any competing interests or state 'None declared': None declared

Please leave your comments for the authors below First, the authors should be congratulated on choosing the approach of PPI in redesigning the provision of healthcare. Patient and Public Involvement is indeed a suitable method to put patient experience at the center of attention within healthcare institutions. As literature shows, there is need for more evidence-based knowledge on this topic and the authors try to improve this body of knowledge. There are however, in my opinion, apparently several (methodological) flaws in the paper. Whether these flaws are a result of actual limitations of the study or a focus on the wrong elements while writing the manuscript is unclear to me. Therefore, my advice is to provide the opportunity for the authors to revise their manuscript and provide more clarity on these issues. If the elements mentioned below are indeed methodological flaws (and can't therefore be amended), I would advise the authors to rewrite their manuscript as a discussion paper, rather than an actual research paper. Again, some insights in this paper could be valuable for future initiatives concerning PPI, even if these are based on experiences instead of rigorous research.

Response: we have addressed and made clearer the aims and objectives (see point 4 and 12, of our response to reviewer 1). This paper presents the results of an evaluation and as such, presents data, although not data gathered in the context of a research study. This paper presents the result of an evaluation that we carried out towards the end and after the FHP had been completed. We have clarified the aims and scope of the paper on pages 5-6 & 7, making it clear that this was an evaluation of the FHP and, therefore the aims of this paper, reporting the evaluation have been clarified. Further, what data and metrics were available to us limited our ability to ascertain the impacts of the PPI initiatives in the development sites.

Background: Although it is useful to receive necessary information on the Future Hospital Program, more information and literature on theoretical models and insights on PPI and co-creation is missing. In the past years many research articles have informed practice concerning these topics. Without referring to international examples of PPI, the authors create the idea that their initiative lacks theoretical input and diminishes transferability. Amongst other examples, I advise the authors to look at initiatives from the U.K., The Netherlands, Germany, France and Belgium. Also, concerning co-creation, a paper by Castro et al. was recently published in Patient Education and Counseling.

Response: (see also response to reviewer 1, point 3). We have contextualised our evaluation within the literature on PPI in HCI on pages 6-7. Given the word limit of the paper, we felt it was difficult to do a survey of initiatives from across Europe – but have added references to general issues raised by this area.

It would also be advisable to make a clear distinction between patient-centred care and patient experiences. I advise the authors to look at the editorial of Dawing and McCormack from 2017. Overall, a more theoretical input is needed in the background section.

Response: (see also response to reviewer 1, point 3). We have clarified how we are using these terms by adding definitions on pages 3, 5-6.

We have included the Future Hospital's definition of patient centred care and noted the conceptual ambiguities of this definition (see page 3). Through-out the paper we have made the distinction between patient-centred care and patient experience clearer.

Methods: The authors state in line 132 that they use a mixed-methods approach to analyze the data. Yet, when reading the paper, it seems that the authors simply have used different methods. Mixed-methods research implies that results from methods are integrated and used in a specific order. I advise to take a close look at: Johnson, R. B., Onwuegbuzie, A. J., & Turner, L. A. (2007). Toward a definition of mixed methods research. *Journal of mixed methods research*, 1(2), 112-133. Moreover, it seems that the quantitative part is missing/not executed in the paper. If so, there is no need of mentioning this in the abstract.

Response: we have changed this to multi-source, to make the point that our evaluation drew on a number of different types of data sources. We have clarified the data sources we have used in the abstract. We have also expanded on the text on the web-based survey, we have added details on the survey in the methods on page 8. It contributed to the analysis in providing additional views on the FHP and we have mentioned some of the relevant findings from this in the results (page 11).

I'm also missing information on ethical consent and ethical approval of this study.

Response: this was included in the original version (page 9).

Results: it is unclear to me which information comes from the analyzed documents, the interviews and are interpretations of the researchers. The result section appears very chaotic and is, at some points, very difficult to follow. In order to provide more structure and make the results more objective, a clearer referral to the data sources is needed.

Response: we have gone through the results in the light of both reviewers' comments and have hopefully made them clearer. We state at the beginning of the methods what data sources we drew on and referenced our use of FHP documentation (page 7-8).

Moreover, it seems strange to me that at this point of the paper, new references are added. What is their goal? If they endorse some statements made, this should be discussed in the discussion section.

Response: there is a tradition in qualitative research that references are included in the results sections, and that also analysis can be incorporated in the results, quantitative conventions of data reporting separate background, results and discussion are of course different. For clarity, we have removed/moved these references.

Also, Table 2 is unnecessary (no added value).

Response: we feel that this provides useful information on the participants, so think that it should be kept in.

Discussion: The discussion section is mostly a reproduction of the results and is writing in a very setting-specific manner. Due to this type of writing, it is difficult to extract insight that can be useful for other initiatives in other countries. My advice is to make the findings more abstract to make them transferrable to other settings. This could be done by crosschecking the finding in the paper with international models on PPI. This would enhance the strength and importance of the paper.

Response: given the scope of the paper, we felt it was important to comment on site specific issues, but we have developed the more general theoretical implications of our work by, explaining our ex-post theory in more detail and linking our findings with the wider literature.

Limitations: In my opinion, one of the major limitations is the underrepresentation of PPI-stakeholders throughout the process (i.e. focusgroups and interviews).

As research has shown before, such underrepresentation affects the role that PPI-stakeholders can/are willing to play and the input they deliver. Please elaborate on this issue. Furthermore, the limitations sections is relatively short in comparison to the other parts of the paper whilst many flaw in the research methodology are present.

Response: we gathered data from all the people who were willing to talk to us, and with our interviews, focus groups, attendance at meetings and drawing on the FHP documentation this covered a good selection of the PPI representatives.

As we have more fully explained the aims of the evaluation we feel that these are not so much flaws in the research methodology as a necessary consequence of the parameters of our evaluation. We have clarified these points in the limitations. We have also critically discussed our method and limitations on page 7, in the outline of our methods.

Conclusion: Based on the comments above, the conclusion will have to be rewritten. I do advice the other to retain phrase 377-to 378 and make this the center of the conclusion. Reading this sentence suddenly gave more meaning to the whole findings and study.

Response: we have revised and rewritten the conclusion and taken on board the comment about the framing of the paper, as one of experiences of moving towards co-production, rather than achieving it.

VERSION 2 – REVIEW

REVIEWER	Dr Helen Barratt University College London
REVIEW RETURNED	12-Jun-2019

GENERAL COMMENTS	Thank you for the opportunity to review this interesting paper a second time. The changes the authors have made have improved the content, but I still have a number of concerns. Major points: 1) Co-production definition a. The authors have inserted a definition of co-production in the introduction, but have chosen to use a definition of co-production in research. There are obviously clear differences in decision-making processes about research and health care delivery. I wonder why they did not use a health care-based definition, for example from the work of either Bate and Robert, or Loeffler? (See for example Batalden M, et al Coproduction of healthcare service BMJ Quality & Safety 2016;25:509-517.)
---

	b. Linked to this, there is a still a disconnect in the way the term coproduction is used in the paper. The authors define what they mean by the expression, but I'm not clear from the results what the research participants meant? Was there a definition given by the RCP? Was this discussed with interviewees? I think it is especially important to be clear about what it meant in the context of the programme (including how this agreed with or differed from established definitions), given the authors' major conclusion that the programme was only 'moving towards' coproduction. 2) I am still unfortunately struggling to get the aims and objectives of the paper clear in my head. Broadly speaking, I think the results section describes the three aims of the PPI programme (I'm not certain where these are identified from), and then goes on to examine in turn the extent to which each of these was met. In the introduction, however, the aim is to 'explore the perceptions and experiences of [stakeholders] of how PPI was operationalised and the benefits and challenges of this work.' In the abstract, the aim is to examine 'the role of PPI... how PPI was operationalised and the benefits and challenges of moving towards coproduction.' All these statements describe slightly different things. Please ensure that the aims/ objectives are stated clearly and consistently throughout the paper, and the results reflect this. 3) The authors have included more information about the evidence behind their assertions in the results section. This is still lacking though from the first section of the results (p11-12). This information about the background to the programme, its rationale, and the assumptions that underpinned it, are really important to understand what follows. Please could you clarify statements such as 'the FHC saw patient experience...' and 'this commitment underpinned FHP's attempt...'. What evidence are they based on? Interviews, focus groups, documentary analysis? If interviews/ focus groups, was there agreement between participants? 4) Linked to this, the authors emphasise the work they have done to elucidate the programme theory that underpinned the intervention, and the importance of this. For the reader unfamiliar with programme theory, it would be helpful to go on and point out where this theory is set out or used. At the moment, the theory isn't really mentioned in the results section, which focuses mainly on empirical findings. This is particularly important to support the claim in the strengths and limitations section (p3) that the theory describes how PPI can be used to improve frontline services. Minor points: 5) The authors have added in information about the timescale of the FHP, which is helpful. It would also be useful to know how long the projects ran for at each site. I may be wrong, but I am guessing it wasn't the full five years. 6) I think the authors undersell the potential value of their work in the first paragraph of the methods section, and elsewhere. Evaluations can absolutely be primary research; the two are not necessarily mutually exclusive. The challenge of arriving on the scene late, when the programme has been running for a while, and decisions about data collection have already been made, is also well acknowledged in the literature. (See for example Raine R, Fitzpatrick R, Barratt H, Bevan G, Black N, et al. Challenges, solutions and future directions in the evaluation of service innovations in health care and public health. Health Services and Delivery Research 2016; 4(16))
--	--

	7) The paper remains quite hard to follow and I had to re-read some sections several times (e.g. description of data collection methods)
--	--

REVIEWER	Simon Malfait Ghent University Hospital
REVIEW RETURNED	03-Jun-2019

GENERAL COMMENTS	Introduction: The authors have provided more background and elaboration to the concept of PPI and co-creation. Also, they have provided more information about the context of this study. By doing so, they have addressed the two main issues from the previous comments. Methods: No further comments Results By restructuring the results section, the paper has become more easily to follow. By providing more quotes and grouping these quotes, the findings of this study are more understandable, leading to interesting insights for the readers. Discussion Overall, the discussion is well organized and very readable. However, I think it is important for future studies to be able to learn from this endeavor. As the authors rightfully state “this is an area that needs further research to develop ways of effectively capturing the impact and effects of PPI in service design and change” (line 505-506), it would therefore be advisable that the authors elaborate on the issues (i.e. limitations) they faced in order to inform these future studies. Please provide the most important lessons learned concerning methodology and how to conduct a similar ‘in vivo’ study on PPI. Conclusion No further comments Overall, the authors should be congratulated on their revisions and the study. Apart from the minor issue concerning the limitations, this study is ready for publication.
--

VERSION 2 – AUTHOR RESPONSE

13-Jun-2019

Reviewer(s)' Comments to Author:

Reviewer: 2

Reviewer Name: Simon Malfait

Institution and Country: Ghent University Hospital Please state any competing interests or state ‘None declared’: None Declared

Please leave your comments for the authors below

Introduction:

The authors have provided more background and elaboration to the concept of PPI and co-creation. Also, they have provided more information about the context of this study. By doing so, they have addressed the two main issues from the previous comments.

Methods:

No further comments

Results

By restructuring the results section, the paper has become more easily to follow. By providing more quotes and grouping these quotes, the findings of this study are more understandable, leading to interesting insights for the readers.

Conclusion

No further comments

Response: we would like to thank the reviewer for their comments that we have clarified the issues as required.

Discussion

Overall, the discussion is well organized and very readable. However, I think it is important for future studies to be able to learn from this endeavor. As the authors rightfully state “this is an area that needs further research to develop ways of effectively capturing the impact and effects of PPI in service design and change” (line 505-506), it would therefore be advisable that the authors elaborate on the issues (i.e. limitations) they faced in order to inform these future studies. Please provide the most important lessons learned concerning methodology and how to conduct a similar ‘in vivo’ study on PPI.

Response: we have added more detail on this on page 19.

Overall, the authors should be congratulated on their revisions and the study. Apart from the minor issue concerning the limitations, this study is ready for publication.

Reviewer: 1

Reviewer Name: Dr Helen Barratt

Institution and Country: University College London Please state any competing interests or state

'None declared': None declared

Please leave your comments for the authors below Thank you for the opportunity to review this interesting paper a second time. The changes the authors have made have improved the content, but I still have a number of concerns.

Major points:

1) Co-production definition

a. The authors have inserted a definition of co-production in the introduction, but have chosen to use a definition of co-production in research. There are obviously clear differences in decision-making processes about research and health care delivery. I wonder why they did not use a health care-based definition, for example from the work of either Bate and Robert, or Loeffler? (See for example Batalden M, et al Coproduction of healthcare service BMJ Quality & Safety 2016;25:509-517.)

Response: we have changed this definition to one specific to service design page 6. We have also refined the element of coproduction we are interested in and referred to Loeffler's work on co-design.

b. Linked to this, there is still a disconnect in the way the term coproduction is used in the paper. The authors define what they mean by the expression, but I'm not clear from the results what the research participants meant? Was there a definition given by the RCP? Was this discussed with interviewees? I think it is especially important to be clear about what it meant in the context of the programme (including how this agreed with or differed from established definitions), given the authors' major conclusion that the programme was only 'moving towards' coproduction.

Response: We have clarified this point, by refining the definition of co-production on page 6. We have included the definition of co-production that the FHP worked with, that focussed on co-design and we have used the term co-production/design to make this clear. The participants talked about co-production and we have used quotes to illustrate their thoughts on this.

2) I am still unfortunately struggling to get the aims and objectives of the paper clear in my head. Broadly speaking, I think the results section describes the three aims of the PPI programme (I'm not certain where these are identified from), and then goes on to examine in turn the extent to which each of these was met. In the introduction, however, the aim is to 'explore the perceptions and experiences

of [stakeholders] of how PPI was operationalised and the benefits and challenges of this work.' In the abstract, the aim is to examine 'the role of PPI... how PPI was operationalised and the benefits and challenges of moving towards coproduction.' All these statements describe slightly different things. Please ensure that the aims/ objectives are stated clearly and consistently throughout the paper, and the results reflect this.

Response: we have clarified the aims, and made the introduction (page 5-6) and abstract consistent. The construction of the goals/aims of the PPI programme are derived from our data collection, FHP documentation and our analysis and how we developed the 'ex-post' theory is clarified on page 10.

3) The authors have included more information about the evidence behind their assertions in the results section. This is still lacking though from the first section of the results (p11-12). This information about the background to the programme, its rationale, and the assumptions that underpinned it, are really important to understand what follows. Please could you clarify statements such as 'the FHC saw patient experience...' and 'this commitment underpinned FHP's attempt...'. What evidence are they based on? Interviews, focus groups, documentary analysis? If interviews/ focus groups, was there agreement between participants?

Response: The first statement quoted came from the FHC report, we have now referenced this to make it clear and the aims of the FHP also came from FHP documentation, and we have also referenced this.

4) Linked to this, the authors emphasise the work they have done to elucidate the programme theory that underpinned the intervention, and the importance of this. For the reader unfamiliar with programme theory, it would be helpful to go on and point out where this theory is set out or used. At the moment, the theory isn't really mentioned in the results section, which focuses mainly on empirical findings. This is particularly important to support the claim in the strengths and limitations section (p3) that the theory describes how PPI can be used to improve frontline services.

Response: we have clarified how we derived the programme theory on page 10 (and see above in terms of clarifying our aims of the paper), and made the link between the theory and how we have used it in reporting results on page 11.

Minor points:

5) The authors have added in information about the timescale of the FHP, which is helpful. It would also be useful to know how long the projects ran for at each site. I may be wrong, but I am guessing it wasn't the full five years.

Response: we have clarified this, in addition to the year that the phases were recruited we have stated that the two phases ran for three and one years respectively with individual teams continuing their quality improvement initiatives beyond the end of the programme (on page 4).

6) I think the authors undersell the potential value of their work in the first paragraph of the methods section, and elsewhere. Evaluations can absolutely be primary research; the two are not necessarily mutually exclusive. The challenge of arriving on the scene late, when the programme has been running for a while, and decisions about data collection have already been made, is also well acknowledged in the literature. (See for example Raine R, Fitzpatrick R, Barratt H, Bevan G, Black N, et al. Challenges, solutions and future directions in the evaluation of service innovations in health care and public health. *Health Services and Delivery Research* 2016; 4(16))

Response: thank you for this point, we have already included this reference and cited it here and rewritten this paragraph on page 7.

7) The paper remains quite hard to follow and I had to re-read some sections several times (e.g. description of data collection methods)

Response: we have gone through the paper, with a particular focus on the data collection section, and edited to make it more readable.